# *Kingella kingae* Spinal Infections in Children

**DOI:** 10.3390/children9050705

**Published:** 2022-05-11

**Authors:** Eleftheria Samara, Nicolas Lutz, Pierre-Yves Zambelli

**Affiliations:** Pediatric Orthopedic Department, Children’s Hospital, Chémin de Montétan 16, 1004 Lausanne, Switzerland; nicolas.lutz@chuv.ch (N.L.); pierre-yves.zambelli@chuv.ch (P.-Y.Z.)

**Keywords:** spinal infections, *Kingella kingae*, Paediatric infection

## Abstract

Nowadays, *Kingella kingae* is considered an important cause of primary spinal infections in children aged between 6 and 48 months. The presentation of the disease is often characterized by mild clinical features and a moderate biological inflammatory response, requiring a high index of suspicion. Performing magnetic resonance imaging (MRI) and obtaining an oropharyngeal specimen and subjecting it to a *K. kingae*-specific nucleic acid amplification test are recommended for its diagnosis. Most patients respond promptly to conservative treatment after administration of antibiotic therapy, which is prolonged for up to 3 months according to the individual clinical and biological response. Invasive surgical procedures are not required except for children who do not improve with antibiotic treatment, develop signs of cord compression, or if the presence of atypical microorganisms is suspected. *Kingella kingae* spinal infections usually run an indolent and benign clinical course, living no permanent sequelae.

## 1. Introduction

Childhood spinal infections represent a continuum of discitis to spondylodiscitis and vertebral osteomyelitis with occasional associated adjacent soft tissue abscesses. Spinal infections are traditionally divided into pyogenic, unspecific granulomatous (tubercular and brucellar infections), and parasitic categories [1]. In developed countries, pyogenic spondylodiscitis is the most frequent form [1]. *Staphylococcus aureus* is isolated in approximately 80% of the cases that occur in the first months of life and in most of those that develop in children older than 48 months [1,2,3,4,5]. Other causative microorganisms are coagulase-negative staphylococci, α-hemolytic streptococci, *Streptococcus pneumoniae*, and Gram-negative rods such as *Escherichia coli* and *Salmonella* spp. [1] In the age group 6 to 48 months, recent studies have suggested that *Kingella kingae* could be the most frequent etiology [6,7,8]. The aim of the present review is to discuss the latest evolutions for *K. kingae* spinal infections, their diagnosis and treatment.

## 2. Epidemiology

The reported incidence for primary pyogenic spinal infections is between 1 and 2 per year per 32,500 pediatric hospital evaluations, or 1 in 250,000 in the general pediatric population [1,2,3,7,9]. A triphasic age distribution in childhood spinal infections has been described and is widely used since it determines the microbiologic epidemiology and the different clinical forms according to the patient’s age [6,7,8]. The neonatal form affects infants less than six months of age and is the most severe form, often presenting with *S. aureus* sepsis and multiple infectious sites. The infantile form affects children aged between six months (the end of maternally derived immunity) and four years, representing the majority of cases of spondylodiscitis in the child population. The third form concerns children older, up to four years, and is usually caused by *S. aureus* [6,7,8]. In the infantile group, *K. kingae* seems to be the most frequent etiology [1,6,8,10,11,12,13,14]. A predominance of incidence in boys has been demonstrated and has been explained by the significantly higher oropharyngeal carriage of the bacterium in boys compared to girls [7,15]. A unique intervertebral disk is usually affected in *K.*
*kingae* spondylodiscitis [12,16]. The lumbar region is the most commonly affected, with decreasing frequency in the thoracolumbar, thoracic, lumbosacral and cervical regions [8,12,14,17,18].

## 3. Pathogenesis

The upper respiratory tract is colonized by many potential bacterial pathogens that reside on the body’s mucosal surfaces, persist, and can spread between individuals [12]. The results of epidemiological investigations on the *K. kingae* carriage have shown increased oropharyngeal colonization in children aged 6 to 29 months, a low colonization rate before the age of 6 months and after 30 months [19,20,21]. This observation, in combination with the peak incidence of primary herpes simplex virus and many respiratory viruses at this age, gives reasonable explication that damage to the mucosal layer caused by a previous or concurrent viral disease predisposes the entry of *K. kingae* into the bloodstream and hematogenous spread to distant sites such as the intervertebral disk [12,22]. In toddlers, the metaphysis of the vertebral body has an abundant vascular bundle, creating an anastomosis with the adjacent metaphyseal vascular bundle through several veins lying in the posterior surface of the disk, penetrating the cartilaginous vertebral plate. Thus, during a *K. kingae* bacteremia event, the microorganism is likely to establish itself in the metaphyseal region of the vertebral body. The bacterium first crosses the cartilaginous vertebral plate, runs through the surface of the disk via the anastomotic veins and subsequently infects the adjacent vertebral metaphysis, finally reaching the disk space between the two involved vertebral bodies [8]. Although there are no blood or lymph vessels in the nucleus pulposus at any age, supply vessels persist in the cartilaginous vertebral endplate until the age of seven, and those in the annulus fibrosus can persist up to the age of 20 years. The obliteration of the rich vascular network that traverses the vertebral endplates cartilage and enters the annulus in young children, coupled with immunological maturity, explain the decreased incidence of the disease after the age of four years.

In advanced or neglected cases, intravertebral, spinal subdural abscesses and paraspinal purulent collections can be formed via dissemination from the adjacent focus [23].

## 4. Clinical and Biological Manifestations

*K. kingae* spinal infection is often characterized by mild clinical presentation. The mean delay between the onset of symptoms and the establishment of a diagnosis and initiating treatment was over one month in 30% of children aged between 6 and 48 months in a recent retrospective, multicenter study [7]. Young children aged 6 to 48 months appear in good general condition, and often present a history of recent or concomitant upper respiratory viral infection. They may present gait disturbances, refuse to sit or walk, cry during diaper changes, and complain of back pain or abdominal pain [24]. Absence of fever at admission or a slightly elevated body temperature are common [7,25,26,27]. The physical examination may reveal localized pain and palpation of the spinous processes; however, it is usually difficult to precisely focalize the most painful region. In advanced cases, physical examination may reveal neurological signs.

*K. kingae* spinal infections are often characterized by a moderate inflammatory response. Most of the patients have normal or near-normal white blood cell counts and C-reactive protein levels [25,28,29,30]. Several models have been described that allow differentiation of osteoarthritic K. kingae infection from those due to other pyogenic pathogens in children under four years of age [31]. Recently, a model to predict a *K. kingae* osteoarticular and thus spinal infection, was proposed and included the following cut-offs for each parameter: age < 43 months, temperature at admission < 37.9 °C, CRP < 32.5 mg/L, platelet count > 361,500/mm^3^ [25].

## 5. Radiologic Findings

Plain X-rays may show a narrowing of the intervertebral space and varying degrees of destruction of adjacent vertebral endplates [8,12]. Scheuerman et al., who studied a cohort of 52 patients with spondylodiscitis, reported that a spinal radiograph was diagnostic in only one patient [32]. Magnetic resonance imaging (MRI) is considered the most sensitive technique for the diagnosis of a spinal infection in the acute phase and is comparable to CT in the chronic stage [7,33,34]. MRI findings are indicative of spinal infection and are not specific to *K. kingae*. Usually, two vertebral bodies and the intervertebral disk are affected. On T1 weighted images, we observe a low signal of the vertebral bodies and the adjacent disk appears with a fluid-like signal intensity, whereas on T2-weighted images, there is a high signal intensity on affected vertebral bodies, with the disk taking on a fluid-like intensity. After administration of gadolinium, contrast in the disk may appear with homogenous enhancement, thick or thin areas of enhancement or patchy areas of enhancement. Fat-suppressed MRI images show an enhancement of affected vertebral body’s’ bone marrow [33,34,35]. Spinal subdural abscesses and intravertebral abscesses are also detected by MRI in protracted or neglected cases [23].

Positron emission tomography with 18 fluorodeoxyglucose (FDG-PET) can distinguish accurately infectious from other inflammatory conditions in the case of MRI unavailability or inconvenience [36].

## 6. Diagnostic Algorithm

High clinical suspicion is required for the early detection of the infection. Every child aged 6 to 48 months admitted with gait disturbances, refusal to sit or walk, crying during diaper changes and who complain of back pain or inexplicable alternatively abdominal pain should be investigated. A history of recent or concomitant upper respiratory viral infection should be searched actively by the clinician. Once clinical suspicion is present, the authors suggest complete inflammatory blood workup (white blood cell counts and platelet counts), erythrocyte sedimentation rate (ESR), C-reactive protein (CRP), blood cultures, and real-time specific assay for *K. kingae* polymerase chain reaction (PCR). Very often, by the time young children present with spinal symptoms, the bacteremic phase has already passed and, thus, blood cultures are usually sterile [37,38,39]. The isolation of *K. kingae* on bacteriological solid media is unsatisfactory even with prolonged incubation [40]. Thus, the use of a real-time specific PCR for *K. kingae* (rtPCR) targeting the *rtx*A, *rtx*B, *cpn*60, or *mdh* genes from oropharyngeal specimens of children aged 6 to 48 months with suspected spinal infections is recommended [12,41].

Performing sensitive *K.*
*kingae*-specific nucleic acid amplification assays (NAAAs) on an oropharyngeal specimen is also essential for a rapid diagnosis. It has been suggested that a positive PCR-specific *K.*
*kingae* on an oropharyngeal specimen in a child with symptomatology compatible with spinal infection provides strong evidence that this microorganism is responsible for spinal infection [7,42]. On the contrary, a negative PCR excludes *K. kingae* as the causative pathogen [42].

We additionally suggest incorporation of the spinal MRI into the diagnostic algorithm of a suspicious acute spinal infection [4,43]. Although MRI imaging has been proved to be a valuable tool to distinguish between *K. kingae* and Gram-positive Cocci non-spinal infections in young children [44], no specific imaging differentiating criteria have been established for spinal infections.

MRI suggestive of spinal infection in combination with a positive PCR assay on a throat specimen can establish the diagnosis, and an immediate intravenous antibiotic treatment should be started before definitive bacteriological results become available.

Biopsies are indicated for patients who do not improve with antibiotic treatment, when signs of cord compression develop, or when clinical suspicion of atypical organisms is suspected [7].

## 7. Treatment and Prognosis

Because of the paucity of *K. kingae* cases and the fact that the etiology is not always confirmed, no *K. kingae*-specific recommendations are available. Despite the absence of a unanimously established treatment protocol, most authors agree on an initial empirical antibiotic treatment with intravenous administration of amoxicillin/clavulanate or cefuroxime in doses 150 mg/kg/jour [7,24]. This therapy is frequently switched to oral antibiotic therapy (usually with a beta-lactam) as soon as improvement of local symptoms and signs of a normalization of body temperature, a decreasing erythrocyte sedimentation rate, and C-reactive protein values occur [7,24]. The duration of antibiotic therapy has varied from 3–12 weeks [24]. This prolonged recommended duration of antibiotherapy is extrapolated from the experience with more aggressive pathogens, especially *S. aureus* and no controlled studies comparing short and long therapies have been conducted to date.

Rest and use of immobilization systems (cast or orthesis) are used as an adjacent therapy [4,45]. There is no conclusive evidence that immobilization of any duration has a beneficial effect on outcome [3]. Most of the patients with spondylodiscitis respond promptly to conservative treatment after administration of antibiotic therapy and they run a benign clinical course [8,16,46]. Persistent residual narrowing of the intervertebral space is uncommon, and the disease leaves no permanent neurologic deficits [8,12]. This can be explained both by the low virulence of *K.*
*kingae* and by its high susceptibility to β-lactam antibiotics [24,47,48].

Invasive procedures, such as aspirations, should be carried out if intervertebral, spinal subdural abscesses or paraspinal purulent collections have been detected by MRI [12,23]. In subdural abscesses, a low threshold for surgical intervention is to be maintained if fever persists, despite adequate antimicrobial treatment or if there are progressive symptoms. Signs of a neurological deficit make surgical drainage and decompression mandatory [8,24]. In epidural abscesses, a laminectomy is appropriate for decompression [49]. Instrumentation with fusion should be suggested in cases where posterior elements are affected, causing instability [50].

## 8. *K. kingae* and Its Antibiotic Susceptibility

*K. kingae* is a facultative anaerobic, Gram-negative bacillus that grows in pairs or chains. This bacterium is a common etiology of bacteraemia and is the leading agent of osteomyelitis and septic arthritis in children aged between 6 and 48 months [13,24]. Apart from spondylodiscitis, soft tissue infections and occult bacteremia are also described in healthy young children. The genus Kingella is represented by the “K” in the acronym HACEK, a group of fastidious Gram-negative bacteria associated with infective endocarditis. Abortive skeletal infections and meningitis secondary to endocardial infection have also been described [12]. K. kingae is usually susceptible to beta-lactam antibiotics that are empirically prescribed to children with suspicion of spinal infection [12,48,51]. Penicillinase stable beta lactams, such as oxacillin, should be precluded from use of confirmed *K.*
*kingae* infections due to the elevated minimal inhibitory concentration in vitro [12,52]. Although beta lactamase as a producer of invasive strains are uncommon and have wide geographic discrepancies [53,54], it is mandatory to use the sensitive nitrocefin method test for beta-lactamase production for all K. kingae isolates from normally sterile body sites [54]. The effect of the beta lactamase enzyme is fully inhibited by clavulanate; thus, as a measure of caution, beta lactamase susceptible penicillins, such as ampicillin, should be avoided and cephalosporin drugs are recommended instead [52,55]. Recently, Keene et al. published high-level flucloxacillin resistance in a *K. kingae* isolate with associated treatment failure of spondylodiscitis. This finding has implications for the choice of empiric therapy in this age group [56]. *K. kingae* invasive isolates are clindamycin non-susceptible, and are highly resistant to glycopeptide antibiotics, which is a serious concern in regions where joint and bone infections caused by community-associated methicillin-resistant *S. aureus* (MRSA) are prevalent and clindamycin or vancomycin is initially prescribed to children with skeletal system infection [12,52,55,57,58]. With rare exceptions, *K. Kingae* is susceptible to aminoglycosides, macrolides, tetracycline, co-trimoxazole, and fluoroquinolones [52,55].

A few studies have demonstrated that the strain of *K. kingae* isolated in the throat was responsible for the osteoarticular infection (spondylodiscitis) [42,59]. Thus, there is a real interest in performing throat swabs both to isolate the bacteria and to study its antimicrobial susceptibility. Currently, we are missing controlled studies that allow evidence-based recommendations to be made on the most effective antibiotic, as well as the necessity or not to investigate β-lactamase production and the optimal duration of treatment for *K. kingae* spinal infections.

## 9. Conclusions

*K. kingae* has emerged as a common cause of pediatric bacteremia and is the leading agent of primary pyogenic spinal infections in children aged 6 to 48 months. The clinical presentation of the disease is often subtle and acute-phase reactants are frequently normal, requiring a high index of suspicion. Oropharyngeal swab PCR detecting a *K. kingae* RTX toxin gene in children aged 6 to 48 months with clinical suspicion of spinal infection is an essential diagnostic tool, since it provides strong evidence that this microorganism is responsible for spinal infection or even stronger evidence that it is not. Moreover, a few studies have demonstrated that the strain of *K.*
*kingae* isolated in the throat swab is responsible for an osteoarticular infection and can be used to study its antimicrobial sensitivity profile. One can suppose that this may be the case in *K.*
*kingae* spinal infections as well. MRI should also be considered as a standard diagnostic tool and invasive procedures, such as biopsies and aspirations, used to detect *K.*
*kingae* and its susceptibility are not recommended. The indirect diagnostic strategy of a spinal infection with the combination of positive oropharyngeal swab and MRI, despite its clinical effectiveness according to the literature, has the limitation of potentially false-positive results and further prospective studies on this subject are recommended.

Beta-lactam antibiotics are administrated empirically for up to 12 weeks and ensure favorable prognosis and seldom lead to long-term sequelae. Nevertheless, the antibiotic of choice and the optimal length of therapy are currently questionable and more studies are needed.

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
