# Peer review of "Kingella kingae Spinal Infections in Children"

_children, 2022, doi:10.3390/children9050705_

Round 1

Reviewer 1 Report

Thank you for the opportunity to review the manuscript entitled Kingella Kingae spinal infections in children: A review.

Major Comments:

Lines 100-105 this is more appropriate in the diagnostic section as opposed to the clinical and biological findings. Please reorganize.

Lines 141-146 please add references for rationale behind spinal MRI recommendation and indicate more clearly if this is author recommended.

Please add a section about the microbiology of K. kingae i.e. a HACEK organism, Gram-stain, other common disease presentations, discovery, virulence factors etc. I think it would be important for a reader to know about the organism itself

Minor Comments

In title kingae should not be capitalized

Line 23: Period should be after the reference. Please correct at other locations in the manuscript as well if able

Line 38: Delete the word currently

Numbers less than 10 please spell out and for numbers greater or equal to 10 please write numerically

Line 48 change comparing to compared

Line 54 change to ---"pathogens that reside on the body’s mucosal surfaces, persist, and can spread from person to person [12].”

The authors alternate between writing Kingella kingae and K. kingae throughout the manuscript. At first mention write the full organism name and the remaining mentions please use K. kingae

Line 50 add in so it reads “with decreasing frequency in the—”

Line 61 change dissemination to hematogenous spread to distant sites

The authors alternate between using disk and disc, please select one and remain consistent.

Line 103 had a hyphen to read real-time PCR, correct throughout manuscript or use the acronym the authors defined.  

Lines 103-105 please indicate in this sentence generally acceptable specimen types for PCR testing. However, it would be best to separate this information into the diagnostic section so please organize.

In the radiological findings section please indicate somewhere that these findings are not specific to K. kingae and perhaps list what would be on the differential diagnosis when relevant in this section

Please define all medical acronyms at first use like ESR, CRP, etc and then remain consistent with acronym use throughout the manuscript

Lines 141- 146 different font and size please correct

Line 134 capitalize K in K. kingae

Line 139 capitalize K in K. kingae, please correct throughout manuscript

Line 146 change subsequently set up to started

Line 160 delete the period after S. aureus

Line 161 change until today to date

Line 169 add a period after K in K. kingae

The authors sometimes use paediatric and pediatric please choose one or the other.

Reviewer 2 Report

Authors present a review on spinal infections in children with Kingella kingae. Epidemology, pathogenesis, radiological findings, diagnostics and treatment options were discussed. However, surgical therapy has only been mentioned but not further discussed. Therefore I suggest to perform a thorough literature review on spinal infections with K.kingae and to include the surgical therapy - it is interesting to see which kind of surgical therapy (decompression, evacuation of empyema or stabilization) in children was needed and when. 

Similar review on musculosceletal infections has appeared, so I suggest to explain what is the additional value of this particular review:

Wong M, Williams N, Cooper C. Systematic Review of Kingella kingae Musculoskeletal Infection in Children: Epidemiology, Impact and Management Strategies. Pediatric Health Med Ther. 2020 Feb 24;11:73-84. doi: 10.2147/PHMT.S217475. PMID: 32158303; PMCID: PMC7048951.

For further Discussion, I suggest to include the following references:

Keene A, Creighton J, Anderson T, Walls T. Kingella kingae Spondylodiscitis: Treatment Failure With Flucloxacillin. Pediatr Infect Dis J. 2022 Jan 1;41(1):48-50. doi: 10.1097/INF.0000000000003357. PMID: 34596625.

A literature review in a form of a table could be useful, since there are less then 30 publications on the subject. 

Reviewer 3 Report

The authors reviewed Kingella Kingae spinal infections in children. The study is intersting, however, I have some concerns to be discussed.

  1. What is the novelty of the current study?
  2.  Please enhance the newer treatment methods.
  3.  Which is the current review, narrative or systmeatic?

I'd like to add the following comments.

The additional comments

1. What is the purpose of this review? Please indicate the goal, for example, to identify new treatments.

2. In the last sentence: you say that more studies are needed. Please show your specific idea of the future study.

3. Please further enrich the discussion by dividing the chapter 7 section into two parts: treatment and prognosis.

Sincerely,

Round 2

Reviewer 2 Report

The authors have sufficiently responded to reviewers remarks.

Author Response

-

Reviewer 3 Report

The authors answered well, so the manuscript is suitable for publication.

Author Response

-